# Increased Radiation Dose Exposure in Thoracic Computed Tomography in Patients with Covid-19

**Massimo Cristofaro, Nicoletta Fusco, Ada Petrone** **, Fabrizio Albarello, Federica Di Stefano** **, Elisa Pianura, Vincenzo Schininà \*, Stefania Ianniello and Paolo Campioni**

Lazzaro Spallanzani, National Institute for Infectious Diseases—IRCCS, Via Portuense, 292, 00149 Rome, Italy; massimo.cristofaro@inmi.it (M.C.); nicoletta.fusco@inmi.it (N.F.); ada.petrone@inmi.it (A.P.); fabrizio.albarello@inmi.it (F.A.); federica.distefano@inmi.it (F.D.S.); elisa.pianura@inmi.it (E.P.); stefania.ianniello@inmi.it (S.I.); paolo.campioni@inmi.it (P.C.)
**\*** Correspondence: vincenzo.schinina@inmi.it

**Simple Summary:** The diagnostic imaging with a chest CT in patients with COVID-19 pneumonia is the key point for early screening, differential diagnosis, staging, the severity of the disease and to plan the possible therapy in the intensive care unit. The evolution of pulmonary changes in this setting requires multiple CT scans in a short period, especially for severe illness. The aim of this study is to assess if there was a variation dose in chest CT scans in COVID-19 patients compared to a cohort with pulmonary infectious diseases at the same time of the previous year to value if there is any modification of exposure dose. We compared 1660 chest CT scans of 597 COVID-19 patients with those of patients hospitalized for infectious respiratory diseases in the same period of the previous year. Our results show that COVID-19 patients are exposed to a higher dose of radiation than other patients, especially in the younger age groups.

**Abstract:** The CT manifestation of COVID-19 patients is now well known and essentially reflects pathological changes in the lungs. Actually, there is insufficient knowledge on the long-term outcomes of this new disease, and several chest CTs might be necessary to evaluate the outcomes. The aim of this study is to evaluate the radiation dose for chest CT scans in COVID-19 patients compared to a cohort with pulmonary infectious diseases at the same time of the previous year to value if there is any modification of exposure dose. The analysis of our data shows an increase in the overall mean dose in COVID-19 patients compared with non-COVID-19 patients. In our results, the higher dose increase occurs in the younger age groups (+86% range 21–30 years and +67% range 31–40 years). Our results show that COVID-19 patients are exposed to a significantly higher dose of ionizing radiation than other patients without COVID infectious lung disease, and especially in younger age groups, although some authors have proposed the use of radiotherapy in these patients, which is yet to be validated. Our study has limitations: the use of one CT machine in a single institute and a limited number of patients.

**Keywords:** COVID-19; diagnostic imaging; chest; CT-scan; radiation dose

## 1. Introduction

A novel coronavirus is currently causing a global outbreak of a respiratory illness termed COVID-19 since December 2019. The exceptional situation, in which we were dealing with a new type of virus whose treatment and therapy were unclear, required improved patient monitoring.

The diagnostic imaging with Computed Tomography (CT) of the chest is the key point for diagnosis in the assessment of early screening, differential diagnosis, the severity of the disease and to plan the possible therapy in the intensive care unit.

Several chest CT findings were reported in more than 70% of RT-PCR test-proven COVID-19 cases, including ground-glass opacities, bilateral distribution, lower lobe in-

volvement, and posterior predilection [1]. A sub-segmental vascular enlargement (more than 3 mm in diameter) in the lung disease was observed [2]. Although in situ thrombosis is certainly a possibility, these findings could be due to hyperemia and increased blood flow. Patients with COVID-19 are at risk of developing thromboembolic complications: the incidence of a pulmonary embolism in patients with COVID-19 who underwent a CT pulmonary angiography ranged between 17% and 35% [1,3–6]. A meta-analysis found that the overall arterial thromboembolism rates of COVID-19 were significantly high. COVID-19 patients who had developed thromboembolism had significantly higher odds of mortality compared to those who did not [7].

The chest CT manifestations of COVID-19 patients are now well known and essentially reflect pathological changes in the lungs. It was reported that COVID-19 could be divided into four stages according to the pathological course of the disease, namely, the early stage (1–4 days), progressive stage (5–8 days), peak stage (9–13 days) and absorption stage (≥14 days). These stages also conform to the characteristics of chest CT images [8].

The evolution of pulmonary changes in this setting requires multiple CT scans in a short period. Therefore, the amount of CT scans range between 3 and 8, according to some authors, and 4 to 8 according to others, in a short period of time, especially for severe illness [9–12]

However, there is insufficient knowledge on the long-term outcomes of this new disease and it may be necessary that patients who contracted the disease undergo several chest CTs for the evaluation of outcomes. This issue is very important for young people who have a longer life expectancy

The aim of this study is to assess if there was a variation dose in chest CT scans in COVID-19 patients compared to a cohort with pulmonary infectious diseases at the same time of the previous year to value if there is any modification of exposure dose.

## 2. Materials and Methods

In our Radiology Department, the chest CT scan protocol has always been evaluated to optimize the patient dose delivered to obtain the best image quality [13–15].

For this purpose, an automated radiation dose monitoring software (DoseWatch, GE Healthcare, Milwaukee, WI, USA) was used for the data collection. DoseWatch is a web-based, cloud-deployed, introductory dose management software to track, analyze and report practice-level data for GE Healthcare CT systems. This software collects radiation dose data directly from a GE Healthcare CT scanner, then summarizes and presents the data via a web application.

The overall dose is obtained from the analysis of CDTI, and DLP values are automatically calculated as an equivalent dose on a phantom (32 cm diameter).

The chest CTs performed in a cohort of COVID-19 positive patients with clinical signs of pneumonia, admitted between March and October 2020, were analyzed to calculate the overall radiological exposure and compared with a group of pneumonia COVID-19 negative patients examined between March and October 2019.

We performed a retrospective study for all patients.

The radiological assessment during hospitalization was performed according to the physician's prescription and internal hospital protocols in relation to the clinical conditions of the patients.

The age, gender, type and number of exams performed, the radiological exposure data in the pre-established period and the outcome of patients were collected. Subsequently, a comparative analysis of the radiation dose in the two groups was performed [16–18].

In both groups, an enhanced CT was performed only in patients with a clinical suspicion of pulmonary embolism.

The data of COVID-19 patients were acquired with a 16-Multi-Slice CT Scanner (Bright Speed General Electric Healthcare, GE Medical Systems, Milwaukee, WI, USA), equipped with an automatic exposure control (AEC) system, noise index (NI) control, and GE IR named ASIR™ (Adaptive Statistical Iterative Reconstruction).

All baseline and follow-up CT scans were performed using 120 kV pp, 250 mA, a pitch of 1.375, thickness of 1.25 mm, tube rotation time of 0.6 s and a scan time of 5 s. The non-contrast scans were reconstructed with slice thicknesses of 0.625 mm and spacing of 0.625 mm with a high-resolution lung algorithm. The CT scan with contrast media, performed in patients with a clinical suspicion of pulmonary embolism, for the chest was reconstructed with a slice thickness of 1.25 mm and spacing of 1 mm. Data from non-COVID-19 patients were acquired using 120 kV pp, 40–140 mA, a pitch of 1.375, thickness of 0.625 mm, tube rotation time of 0.5 s and a scan time of 8 s. The images were subsequently reconstructed in a thickness of 0.625 mm for contiguous axial slices.

The use of a higher current (250 mA) in COVID-19 patients was necessary to minimize acquisition time and reduce respiratory movement artifacts, always present in COVID-19 patients with pneumonia and severe dyspnea.

For the dose calculation, we evaluated the value of DLP and CTI, obtained directly from the CT scanner.

All studies were stored and displayed on a picture archiving and communication system workstation (Impax ver. 6.6.0.145, AGFA Gevaert SpA, Mortsel, Belgium).

All analyses were performed using a specific freeware software (StatPlus: Mac by AnalystSofte) and IBM SPSS Statistics (version 26.0, Armonk, NY, USA, IBM, Corp).

We evaluated the number of chest CTs in two study groups (March–October 2019–2020) with an Analysis of Variance (ANOVA).

The study was conducted within the Recoveri Project, which was approved by the Ethics Committee (decision n. 164/2020). Informed consent was obtained from all subjects involved in the study.

## 3. Results

One thousand six hundred and sixty chest CT scans, with and without contrast media, of 597 COVID-19 patients admitted in our hospital were acquired from 1 March to 31 October 2020.

The mean age of patients included was 64.9 years, 70.1% were men; 22.55% (187/980) of patients underwent a maximum of 10 examinations.

The CT scans were performed with 140 mAs, 1.4 pitch, 0.35 s rotation time, and a 2.3 s mean scan time in COVID-19 patients.

The mean total cumulative effective dose of lung CTs in all COVID-19 patients was 11.03 mSv (median 9.04 mSv, SD 5.81). Women received more effective doses than men (mean 11.52 mSv vs. 10.81 mSv; Tables 1 and 2).

A total of 59 out of 980 people (6.02%) received cumulative doses between 20 and 30 mSv, and 22 people (2.24%) received doses greater than 30 mSv (maximum 31.24 mSv).

**Table 1.** Dose in COVID patients by age and gender.

| Age, Gender and Dose in COVID-19 Patients | | | | | | | |
| --- | --- | --- | --- | --- | --- | --- | --- |
| **Age** | **Male (no)** | **Male (%)** | **Female (no)** | **Female (%)** | **% TOT** | **Mean (mSv)** | **Median (mSv)** |
| 21–30 | 13 | 59.09 | 9 | 40.91 | 2.29 | 11.01 | 7.86 |
| 31–40 | 52 | 74.29 | 18 | 25.71 | 7.29 | 11.80 | 9.71 |
| 41–50 | 120 | 82.19 | 26 | 17.81 | 15.21 | 11.01 | 8.95 |
| 51–60 | 212 | 75.18 | 70 | 24.82 | 29.38 | 11.71 | 9.97 |
| 61–70 | 118 | 71.08 | 48 | 28.92 | 17.29 | 11.31 | 9.19 |
| 71–80 | 108 | 62.79 | 64 | 37.21 | 17.92 | 11.12 | 8.64 |
| 81–90 | 48 | 52.17 | 44 | 47.83 | 9.58 | 10.33 | 7.64 |
| >90 | 2 | 20.00 | 8 | 80.00 | 1.04 | 10.91 | 9.99 |

The number of examinations normalized for age showed: (a) patients between 21 and 30 years (22, 2.29%) received a total of 38 CT scans (average 1.72); (b) 31–40 years (70, 7.29%) 129 CT scans (average 2.74), (c) 41–50 years (146, 15.71%) 227 CT scans (average 2.58), (d) 51–60 years (282, 29.38%) 476 CT scans (average 3.05), (e) 61–70 years (166, 17.29%)

335 CT scans (average 2.84), (f) 71–80 years (172, 17.92%) 311 CT scans (average 3.2), (g) 81–90 years (92, 9.58%) 179 CT scans (average 2.39) and (h) over 91 years (10, 1.04%) 11 CT scans (average 1.58).

**Table 2.** Dose in no COVID patients by age and gender.

| Age, Gender and Dose in Patients 2019 | | | | | | | |
|---|---|---|---|---|---|---|---|
| Age | Male (no) | Male (%) | Female (no) | Female (%) | % TOT | Mean (mSv) | Median (mSv) |
| <20 | 5 | 71.43 | 2 | 28.57 | 0.73 | 5.72 | 3.45 |
| 21–30 | 37 | 68.52 | 17 | 31.48 | 5.63 | 5.90 | 5.38 |
| 31–40 | 64 | 71.91 | 25 | 28.09 | 9.27 | 7.03 | 5.40 |
| 41–50 | 77 | 60.63 | 50 | 39.37 | 13.23 | 7.19 | 6.01 |
| 51–60 | 176 | 67.95 | 83 | 32.05 | 26.98 | 8.62 | 7.38 |
| 61–70 | 135 | 67.84 | 64 | 32.16 | 20.73 | 7.41 | 6.58 |
| 71–80 | 97 | 57.74 | 71 | 42.26 | 17.50 | 8.20 | 8.00 |
| 81–90 | 47 | 65.28 | 25 | 34.72 | 7.50 | 7.78 | 7.14 |
| >90 | 0 | 0 | 3 | 100 | 0.31 | 8.05 | 6.68 |

The chest CT scan doses of COVID-19 patients were compared with patients hospitalized for infectious respiratory diseases in the same period of the previous year (March–October 2019) when 1919 CT scans in 978 patients with infectious respiratory diseases were performed; 19% were patients admitted to the intensive care, compared with 7.5% the previous year.

The mean age at inclusion was 60.9 years, 66.7% were men and 28.7% of the patients performed one lung CT scan, with a maximum of three examinations (Table 3).

**Table 3.** Comparison between doses in COVID-19 and non-COVID-19 patients by age group.

| Dose Per Person Exposed in COVID-19 Patients and Control Group | | | | | | | | |
|---|---|---|---|---|---|---|---|---|
| COVID-19 Patients | | | | No COVID Patients (2019) | | | | |
| Age | Mean (mSv) | Median (mSv) | Standard Deviation | 90 Percent | Mean (mSv) | Median (mSv) | Standard Deviation | 90 Percent | *p* Value |
| 21–30 | 11.01 | 7.86 | 7.29 | 5.38 | 5.90 | 5.38 | 3.78 | 1.53 | 0.022 |
| 31–40 | 11.80 | 9.71 | 9.71 | 5.43 | 7.03 | 5.40 | 4.40 | 2.55 | 0.051 |
| 41–50 | 11.01 | 8.95 | 6.41 | 5.20 | 7.19 | 6.01 | 4.57 | 2.51 | 0.001 |
| 51–60 | 11.71 | 9.97 | 5.66 | 5.40 | 8.62 | 7.38 | 4.75 | 3.31 | 0.042 |
| 61–70 | 11.31 | 9.19 | 6.07 | 5.38 | 7.43 | 6.58 | 3.75 | 3.35 | 0.001 |
| 71–80 | 11.12 | 8.64 | 4.40 | 5.38 | 8.20 | 8.00 | 3.84 | 3.40 | 0.001 |
| 81–90 | 10.33 | 7.64 | 6.23 | 4.31 | 7.78 | 7.14 | 3.61 | 3.42 | 0.002 |
| >90 | 10.91 | 9.99 | 5.18 | 10.73 | 8.05 | 6.68 | 3.53 | 5.38 | 0.001 |

The mean total cumulative effective dose in the lung CTs of all 2019 patients group was 7.42 mSv. However, 18 out of 979 (1.83%) patients from the 2019 group received doses greater than 20 mSv (maximum 21.53 mSv).

The dose in COVID-19 patients increased from 32.77% (10.33 mSv vs. 7.78 mSv; age range 81–90) to 86.61% (11.01 mSv vs. 5.90 mSv; age range 21–30; Figure 1).

Each COVID-19 patient received an average of 2.78 CT examinations (min 1, max 10), while the non-COVID-19 control group received an average of 1.1 CT scans each (min 1, max 3).

The relationship between the number of CT scans performed and the outcome within the 2020 group of patients resulted as follows: Mortality affected 12.4% of patients. Among these, 26.7% (33/124) did not receive any CT scans, 45.1% (56/124) received only one CT scan, 11.3% (14/124) received two CT scans, 10.5% (13/124) received three CT scans, 4.8% (6/124) received four CT scans, 0.8% (1/124) received five and ten CT scans, respectively (Figure 2).

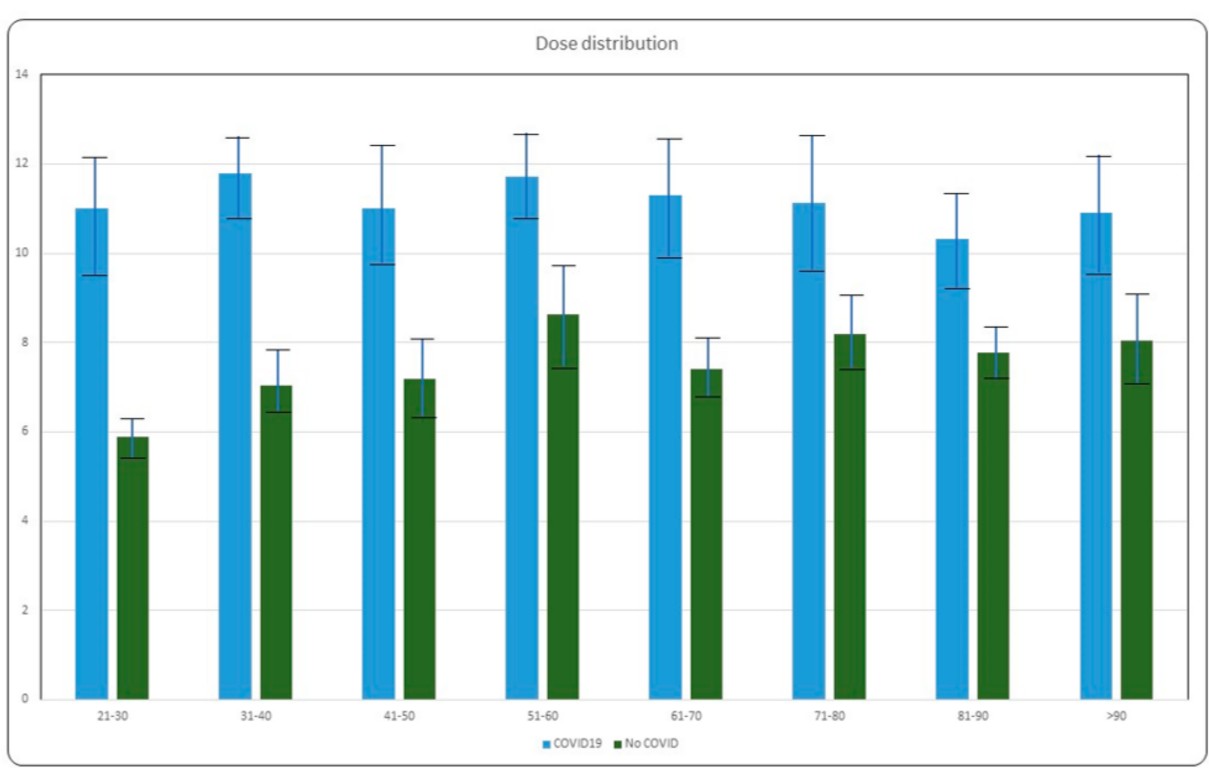

**Figure 1.** Comparison of dose distribution in COVID-19 and no COVID patients by age group.

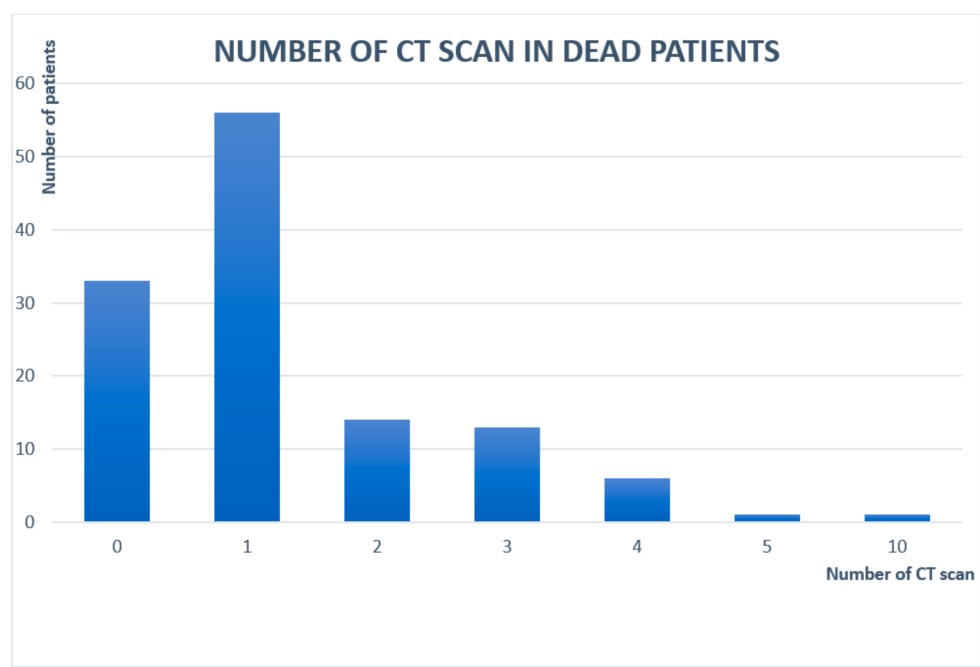

**Figure 2.** Number of CT scans performed in patients who died from COVID-19.

## 4. Discussion

According to the Fleischner Society [19], chest imaging "is not indicated" in patients with the suspected COVID-19 disease but with mild clinical features. The statement supports the use of imaging in COVID-19 patients with a worsening respiratory status as well as in those with suspected COVID-19 and moderate to severe presentation with a high pre-test probability of infection. On the other hand, CT scans in patients with suspected or known COVID-19 infections could be justified in certain cases [20].

The International Commission on Radiological Protection (ICRP) on COVID-19 and chest CT "Protocol and dose optimization" found that non-contrast phase lung examination was obtained (86%, 103/120), although others also acquire one or two phases with contrast media. The chest CT's most used protocol in COVID-19 patients was associated with the same dose as routine scans (55%, 64/117; 5–10 mGy), whereas low-dose (43%, 50/117; <5 mGy) and high-dose (3%, 3/117; >10 mGy) CT protocols were used at the remaining sites [21,22].

Exposure to radiation during CT scans should be carefully monitored, especially in young people that have a longer life expectancy, so it may have remote consequences.

Although a single CT scan does not represent a significant risk to the patients' health, repeated CT scan over a short time interval could increase the biological damage.

A recent study shows that the standard chest CT study performed without contrast means that the standard-dose chest CT (5 mSv) was associated with DNA double-strand breaks (before vs. after CT, 0.11 vs. 0.16 per cell, respectively; $p < 0.001$) and chromosome aberrations (before vs. after CT, 7.6 vs. 9.7 per 1000 metaphases, respectively; $p = 0.003$) [23].

Nevertheless, the recently published monographs on epidemiological studies of low dose ionizing radiation and cancer risk report positive excess relative risk (ERR) at 100 mGy for adults [24].

There are few studies on the evaluation of CT protocol in patients with known or suspected COVID-19 pneumonia. Kang et al. reported adequate assessment of pulmonary opacities related to COVID-19 pneumonia at 100 kV with a tin filter (spectral shaping filter, Siemens Healthineers, city, Enlargen, Germany) and iterative reconstruction technique with a volume CT dose index (CTDI vol) of 0.4 mGy versus standard-dose protocol at 3.4 mGy. Another study applied 100 kV with a tin filter and a 0.6 s exposure time using a high pitch and fast gantry rotation time to acquire chest CT examinations in COVID-19 pneumonia at 0.6 mGy CTDIvol, which were comparable to chest CT at 6.4 mGy [9,11].

The number of CT scans ranges from 3 to 6 according to some authors and 4 to 8 according to others, in a short period of time, especially for severe illness [9–12]. Mohammad reported the mean time between the CT scans in different stages of the disease: mild (5.8 ± 1.2 days), common (4.9 ± 1.4 days), severe (4.5 ± 1.0 days) and critical (3.7 ± 1.1 days).

Therefore, in the current global pandemic, there have been patients undergoing routine chest CT examinations of up to 14 CT scans in the follow up of the disease within 20 days from 21 to 67 mSv, which the latter has up to 3.5 times the allowable dose as per the recommendations set by the International Commission on Radiological Protection (ICRP), Report 103 [25].

A lack of guidance on how to best use a CT with COVID-19 has led to significant variations in standard measurements of CT radiation doses. To assess how CT has been used in the pandemic, Homayounieh and colleagues used data from an International Atomic Energy Agency (IAEA) survey conducted between May and July that assessed use, protocols, and radiation doses of CT exams performed for COVID-19 [26].

The team found eight-fold variations in median CTDI vol (2–17 mGy) and 10-fold variations in median DLP (76–786 mGy-cm) with an effective dose, 1–8 mSv. Additionally, some continents showed wide ranges: For example, cumulative DLPs for Latin American patients were 503 mGy, compared with a range of 306 mGy to 382 mGy for the three other continents [26].

About 30% of the patients ($n = 225$) underwent two to eight chest CT examinations in less than a month.

"Considerable variations in scan protocols are observed in healthcare sites worldwide and the current level of radiation dose is much greater than that of the proposed low-dose CT scan protocols", wrote Choonsik Lee. "It will be crucial for future studies on CT dose during COVID-19 follow-up to and to assess the risks and benefits of follow-up CT scans in the context of COVID-19" [27].

On the other hand, low-dose radiotherapy has been considered as one of the potential treatments for COVID-19 pneumonia. The results of the pilot trial evaluating low-dose whole-lung irradiation (LD-WLI) in patients with COVID-19 pneumonia show the feasibility of treatment in patients with moderate COVID-19 pneumonia [28–31].

The analysis of our data shows an increase in the overall mean dose in COVID-positive patients compared with COVID-negative patients of approximately 48.65% (11.03 mSv vs. 7.42 mSv).

In COVID-19 positive patients, there are minimal variations in the mean dose of various age groups (11 mSv). On the contrary, in non-COVID-19 patients, there are variations in doses, with a progressive increase in doses with increasing age from 5 to 8.5 mSv.

Our standard protocol in chest CT scans for non-COVID-19 pneumonia is a low dose, especially in young people; in the COVID-19 patients, we used a scan protocol to minimize respiratory artifacts dedicated to dyspnoic or ventilator assisted patients.

Our study has limitations: the use of a single CT scan and a limited number of patients. On the other hand, the percentage increase in absorbed dose appears to be more relevant for us than the absolute values of the absorbed dose (machine-dependent).

## 5. Conclusions

In our institution, in the first phase of the pandemic (one month), only CT scans without contrast medium was performed. Subsequently, the suspicion of pulmonary thromboembolism or micro thrombosis and the presence of other pathologies in patients (neoplasms in particular) led to a marked increase in CT examinations with contrast medium to monitor vascular and parenchymal pathology.

The comparison between the two groups of age shows how the higher dose increase occurs in the younger age groups 21–30 and 31–40 years and in the 81–90-year-old group. Our results show that COVID-19 positive patients are exposed to a significantly higher dose of ionizing radiation than patients with COVID-negative infectious lung diseases, and especially in the younger age groups.

This aspect must be considered for all patients with this new disease that is still so little known, especially in a follow-up to evaluate any outcomes [31].

**Author Contributions:** M.C., N.F. and A.P. have contributed equality to conducting a literature review, F.A., F.D.S. and E.P. performed the writing, and V.S., S.I. and P.C. edited the article. All authors have read and agreed to the published version of the manuscript.

**Funding:** This research received no external funding.

**Institutional Review Board Statement:** The study was conducted according to the guidelines of the Declaration of Helsinki, and approved by the Ethics Committee of National Institute for Infectious Disease Lazzaro Spallanzani, Rome, Italy n. 235 (12/29/2020).

**Informed Consent Statement:** Informed consent was obtained from all subjects involved in the study.

**Data Availability Statement:** All data were collected anonymously in our archives.

**Acknowledgments:** The authors are grateful to the *COVID-19 team of Radiology Department at the Lazzaro Spallanzani National Institute of Infectious Diseases, Rome*, Italy, for providing daily support in this critical situation: Angela Aprea, Ivan Berretta, Nicola Caretto, Eugenio Celli, Marco Cellini, Felicia Coroian, Mirko Costa, Daniele Di Bartolomeo, Luigia Di Felice, Tiziana Fazio, Cristian Gaglio, Filomena Graziano, Chiara Iapalucci, Maria Rosella Longo, Luisa Mari, Roberto Mengarelli, Maurizio Morea, Daniel Panzica, Valter Possanzini, Nicola Soccio, Cristina Stornaiuolo, Maruska Tabacco, Natalia Tartaglia, Alessia Trombetti and *COVID-19 INMI Study Group* Maria Alessandra Abbonizio, Amina Abdeddaim, Chiara Agrati, Fabrizio Albarello, Gioia Amadei, Alessandra Amendola, Mario Antonini, Tommaso Ascoli Bartoli, Francesco Baldini, Raffaella Barbaro, Barbara Bartolini, Rita Bellagamba, Martina Benigni, Nazario Bevilacqua, Gianlugi Biava, Michele Bibas, Licia Bordi, Veronica Bordoni, Evangelo Boumis, Marta Branca, Donatella Busso, Marta Camici, Paolo Campioni, Maria Rosaria Capobianchi, Alessandro Capone, Cinzia Caporale, Emanuela Caraffa, Ilaria Caravella, Fabrizio Carletti, Concetta Castilletti, Adriana Cataldo, Stefano Cerilli, Carlotta Cerva, Roberta

Chiappini, Pierangelo Chinello, Carmine Ciaralli, Stefania Cicalini, Francesca Colavita, Angela Corpolongo, Massimo Cristofaro, Salvatore Curiale, Alessandra D'Abramo, Cristina Dantimi, Alessia De Angelis, Giada De Angelis, Maria Grazia De Palo, Federico De Zottis, Virginia Di Bari, Rachele Di Lorenzo, Federica Di Stefano, Gianpiero D'Offizi, Davide Donno, Francesca Faraglia, Federica Ferraro, Lorena Fiorentini, Andrea Frustaci, Matteo Fusetti, Vincenzo Galati, Roberta Gagliardini, Paola Gallì, Gabriele Garotto, Saba Gebremeskel Tekle, Maria Letizia Giancola, Filippo Giansante, Emanuela Giombini, Guido Granata, Maria Cristina Greci, Elisabetta Grilli, Susanna Grisetti, Gina Gualano, Fabio Iacomi, Giuseppina Iannicelli, Stefania Ianniello, Giuseppe Ippolito, Eleonora Lalle, Simone Lanini, Daniele Lapa, Luciana Lepore, Raffaella Libertone, Raffaella Lionetti, Giuseppina Liuzzi, Laura Loiacono, Andrea Lucia, Franco Lufrani, Manuela Macchione, Gaetano Maffongelli, Alessandra Marani, Luisa Marchioni, Andrea Mariano, Maria Cristina Marini, Micaela Maritti, Alessandra Mastrobattista, Giulia Matusali, Valentina Mazzotta, Paola Mencarini, Silvia Meschi, Francesco Messina, Annalisa Mondi, Marzia Montalbano, Chiara Montaldo, Silvia Mosti, Silvia Murachelli, Maria Musso, Emanuele Nicastri, Pasquale Noto, Roberto Noto, Alessandra Oliva, Sandrine Ottou, Claudia Palazzolo, Emanuele Pallini, Fabrizio Palmieri, Carlo Pareo, Virgilio Passeri, Federico Pelliccioni, Antonella Petrecchia, Ada Petrone, Nicola Petrosillo, Elisa Pianura, Carmela Pinnetti, Maria Pisciotta, Silvia Pittalis, Agostina Pontarelli, Costanza Proietti, Vincenzo Puro, Paolo Migliorisi Ramazzini, Alessia Rianda, Gabriele Rinonapoli, Silvia Rosati, Martina Rueca, Alessandra Sacchi, Alessandro Sampaolesi, Francesco Sanasi, Carmen Santagata, Alessandra Scarabello, Silvana Scarcia, Vincenzo Schininà, Paola Scognamiglio, Laura Scorzolini, Giulia Stazi, Fabrizio Taglietti, Chiara Taibi, Roberto Tonnarini, Simone Topino, Francesco Vaia, Francesco Vairo, Maria Beatrice Valli, Alessandra Vergori, Laura Vincenzi, Ubaldo Visco-Comandini, Pietro Vittozzi, and Mauro Zaccarelli.

**Conflicts of Interest:** The authors certify that there is no conflict of interest with any financial organization regarding the material discussed in the manuscript.

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
