# Peer review of "Increased Radiation Dose Exposure in Thoracic Computed Tomography in Patients with Covid-19"

_radiation, doi:10.3390/radiation1020014_

Round 1

Reviewer 1 Report

Thank you for an interesting manuscript describing the increased radiation exposure by CT scan of COVID-19 patients compared with other pulmonary patients. I have both some general points and some specific points.

General:

The English grammar and choice of words of the manuscript has to be improved, as many sentences are hard to read and some paragraph almost unintelligible.

Consistency is lacking:

  • "COVID-19" or "COVID 19" or "COVID19"
  • You switch between using "," and "." for decimals
  • Be clear if "CT scans" and "CT examinations" are the same thing, or if there is a difference? (I.e. can an examination contain multiple scans?)

Generally in many statements it is unclear, if they descrive the COVID 19-cohort or the total patient cohort. And for many of the counts (e.g. the 980 scans in line 94) it is unclear which cohort these relate to.

Moreover, why did you not (also) include non-COVID patients from 2020 in the study?

Major specific points:

  • Line 75: Yous state that all scans were performed with 250 mA, but later you write that the non-COVID scans were performed with 40-140 mA. Which is correct? And if the 250 mA is for the COVID patients only, why is there a difference? Furthermore, what do "baseline" and "followup" refer to?
  • Line 87-88: Did you test normality assumptions (and other relevant model assumptions) of your linear regression models? It seems quite improbable that residuals for a count of 1-10 scans per person should be normally distributed.
  • Line 87-88 and Tabel 3: How did you compare the doses between groups? You do not specify any statistical approach for the p-values reported in Table 3.
  • Table 3 repeats a lot of results from Tabel 1 and 2.
  • Figure 2 does give very little information, it should include comparision with surviving patients to make sense.
  • Line 204-206: This part sounds like a description of your methods and not a discussion.

Minor specific points:

  • Line 103: What does "normalized" mean?
  • Line 103-109: Consider adding these numbers to Table 1/2 instead of listing them as text.
  • Line 116: Table 3 does not show information related to this sentence.

Author Response

The new manuscript  is complete with the required corrections

Reviewer 2 Report

Review Radiation-1180084

Summary
This study aimed to evaluate the possibility of an increase in radiation dose exposure in chest CTs in Covid-19 patients compared to patients with pulmonary infectious disease. The study showed that patients suffering from Covid-19 had a significant increase in radiation dose exposure, especially in young adults.

Title
The title is confusing to me.

Is the entire title meant to be a question or only the second part? Furthermore, as far as I understand the study, you are ‘only’ evaluating the radiation dose exposure and not the therapeutical effect of radiation. I would suggest changing the title to something like:  

Increased radiation dose exposure in thoracic computed tomography in patients with Covid-19

Key words and abbreviations
Ok.

Informed Consent
Was informed consent waived by the IRB in the Recoveri Project? (line 89/90)

Ethical Committee (if relevant)
Ok

General:

Covid-19 is sometimes spelled with a hyphen (-) and sometimes without. Please stay consistent throughout the manuscript.

The entire manuscript needs to be proofread by a native speaker. It has numerous grammatical errors. I won’t point out all of them.

Was this study performed retrospective or prospective? Since the pneumonia patients are from 2019 probably retrospective but what about the Covid-19 patients? Please state this clearly in MM

Introduction

Line 25: Coronaviruses are present for a while (e.g. SARS-CoV-1), thus, Covid-19 is NOT a completely new pathogen but rather a new type.

Line 28: You may delete ‘staging’ from the list. Staging is performed in patients with tumor, which is included in differential diagnosis.

Line 39: …who had …thromboembolism had significant ….

May I suggest rearranging the introduction. Right now you start with the novelty of the virus, before talking about CT, then the complications of the disease and then coming back to CT. Currently the introduction is ‘bumpy’ to read’. Try to group the information about CT together.

Line 47: … the amount of CT scans range between 3 and 8…

Material and Methods
Line 61-64: Were the patients included consecutively?  Were there any inclusion or exclusion criteria for Covid-19 as well as pneumonia patients?

Line 68: … number of examS …

Line 76-79: Which criteria determined if exams were conducted with and without contrast? The usual, such as allergy, renal function, … or were other factors?

Please describe all dose values you evaluated.

Results
Line 95:  Delete ‘to’ from the sentence

Line 101: Should the ‘e’ be an ‘and’ or a dash?

Why did you decide not to evaluate DLP and CTDIvol? These are relevant values to determine the radiation dose exposure. Not comparing these values results in a major limitation to your project.

Discussion
Line 137: First and only time you call the virus ‘SARS-CoV-2’. Stay consistent or at least mention the actual name first time you mention Covid-19.

Line 142 – 145: This sentence is not clear to me. What do you try to say?

Line 149: delete the ‘the’ at the end of the row

Conclusion:

The conclusion should not have any results, such as percentage and no references to Tables in it but simply conclude your results.

The limitations should be at the end of the discussion.

References
Ok.

Figures and Tables
Ok.

--------------------------------------------------------------------------------------------------------------------------------------

Dear Ms. Zhang,

I just finished the review concerning the manuscript Radiation-1180084 entitled " Increases Radiation Dose for Chest Scan in COVID-19 Patients: Damage or Therapy?".

Statement:
Although the presented study has topic which is highly discussed, the scientific soundness and scientific presentation of this particular research project is questionable. In my opinion, the manuscript should be rejected for publication in ‘Radiation’.

Major Strengths:

  • More or less relevant topic due to the ongoing pandemic, especially since every hospital has their own management.

Major Weaknesses:

  • Manuscript is badly structured and not presented in a scientifical manor (i.e. there are no in- or exclusion criteria mentioned for patients, …).
  • Simply comparing the cumulative dose and not evaluating the DLP and CTDI is a major limitation to the study

Minor Weaknesses

  • English language. Manuscript need to be proofread by a native speaker

Sincerely,

Sven S. Walter

Author Response

Thanks for comments.

We have corrected the manuscript in all the points indicated and explained better the unclear points.

Reviewer 3 Report

This document provides the increase of effective dose from chest CT in COVID-19 patients compared with the patients in 2019. The latest information will be useful to optimization of medical exposure although these are limited data in a single institute. More basic information about the irradiation conditions of chest CT should be provided for the purpose of this document.

Specific comments

1) In Line 19, “a single CT scan” must be “one CT machine of a single institute”.

2) Calculation of effective dose is done by dose monitoring software, DoseWatch. No information on DoseWatch is provided. Some references should be provided for readers who do not have the sufficient information.

3) In Tables 1 and 2, the averages and SDs of effective dose, mA, pitch, thickness and rotation time should be included. These information would be more useful to more consideration on optimization.

4) The effective doses from chest CT in COVID patients seem to vary widely, although 250mA, etc are the same among the patients. The reason of wide variation should be discussed.

5) Statistical analysis uses a univariate linear regression model. No results are there. The p values show in Table 3. The statistical method must be described like t-test or ANOVA.

6) Figure 1 is a key graph in the document. Each bars must be shown with Standard error of the average doses.

7) In Line 136,  the reference of the Fleischner society should be provided.

8) In Line 142, ICRP has addressed no statement about CT optimization of COVID-19, and only recommends justification and optimization of medical exposure by Pub 102, Managing patient dose in multi-detector computed tomography and Pub 135, Diagnostic reference levels in medical imaging.

9) In Line 175, incorrect message is there. ICRP has not addressed the allowable dose. Misunderstanding is recognized on the dose limits to occupational exposure.

10) In References, there is an inconsistent description of journals. Check all the references and follow the author guideline.

Author Response

Thanks for revision.

We have appreciated and made all the changes reported.

Round 2

Reviewer 1 Report

Thank you for the modified version of the manuscript. The changes have made many points of you paper much more clear.

I still would suggest improving the English further, but it has imporved considerable.

Author Response

Thanks.

Reviewer 2 Report

I comment the authors to the manuscript

Author Response

Thanks